# A Preliminary Study on the Resolution of Electro-Thermal Multi-Physics Coupling Problem Using Physics-Informed Neural Network (PINN)

**Yaoyao Ma** [1,2,3] **, Xiaoyu Xu** [4] **, Shuai Yan** [4] **and Zhuoxiang Ren** [4,5,*]

1. Institute of Microelectronics of Chinese Academy of Sciences, Beijing 100029, China; mayaoyao@ime.ac.cn
2. University of Chinese Academy of Sciences, Beijing 100049, China
3. Beijing Key Laboratory of Three-Dimensional and Nanometer Integrated Circuit Design Automation Technology, Beijing 100029, China
4. Institute of Electrical Engineering, Chinese Academy of Sciences, Beijing 100190, China; xuxiaoyu@mail.iee.ac.cn (X.X.); yanshuai@mail.iee.ac.cn (S.Y.)
5. Group of Electrical and Electronic Engineering of Paris, Sorbonne Université, Université Paris-Saclay, CentraleSupélec, CNRS, 75005 Paris, France
* Correspondence: zhuoxiang.ren@upmc.fr

**Abstract:** The problem of electro-thermal coupling is widely present in the integrated circuit (IC). The accuracy and efficiency of traditional solution methods, such as the finite element method (FEM), are tightly related to the quality and density of mesh construction. Recently, PINN (physics-informed neural network) was proposed as a method for solving differential equations. This method is mesh free and generalizes the process of solving PDEs regardless of the equations' structure. Therefore, an experiment is conducted to explore the feasibility of PINN in solving electro-thermal coupling problems, which include the electrokinetic field and steady-state thermal field. We utilize two neural networks in the form of sequential training to approximate the electric field and the thermal field, respectively. The experimental results show that PINN provides good accuracy in solving electro-thermal coupling problems.

**Keywords:** electro-thermal coupling; deep learning; physics-informed neural network; PDEs



## 1. Introduction

The resolution of multi-physics problems comes down to the computation of partial differential equation (PDE) solutions. General numerical methods for handling PDEs include the finite element method (FEM), the finite difference method (FDM), the finite volume method (FVM), etc. All these methods require a discrete representations of the domains and utilize interpolation functions to obtain the solution not on the discrete set. The discretization represents the domain well for low-dimensional problems, but not for high-dimensional ones, as the number of elements increases exponentially with the dimensionality. In addition, these methods only solve the PDEs at discrete points, and thus require interpolation or slope behavior for other points or other fields; this property makes the solution of the state variables at the interpolated points less accurate [1], particularly the derivatives of the state variables.

Deep learning has recently achieved great success in the fields of science and business [2–4]. Due to these advances, many scientists have been working to embrace deep learning in the computation of physical problems. Most of these studies are data driven [5,6], which learn certain corresponding relationships between the input data and the output data, and then output corresponding predictions for the new input data. The correspondence is usually unknown and not obvious; hence, it requires a large amount of training data to learn, and that may be arduous. Furthermore, the training process is usually more like a "black box," and the hidden layers' physical meanings are still unknown, making the network training time consuming. To alleviate this drawback, loss functions, which are

used to determine the error between the output of neural network and a given target value, are embedded with some physical constraints. The deep learning training process could be directed to minimize the governing equations' residual. Physical constraints can compensate for the lack of data so that less training data are needed to approximate the solution of the equation. This idea of physically constrained deep learning dates back to the 1990s [7–9]. Owing to NVIDIA's breakthrough in GPU computing and the perfection of deep learning theory, this idea was recently revived. Sirignano et al. [10] proposed a mesh-free deep Galerkin method (DGM) based on deep learning, and solved high-dimensional PDEs, including the Hamilton–Jacobi–Belllman PDE and the Burgers' equation. Wang et al. [11] and He et al. [12] leveraged deep neural networks to solve interface problems in domains with multiple materials. Berg et al. [13] focused on the approximation of PDEs with complex geometries. Ovcharenko et al. [14] utilized deep learning to predict low frequency seismic data. Sun et al. [15] proposed a ML-decent method to learn the optimization algorithm and applied it to full-waveform inversion problem.

Recently, Raissi et al. [16] formulated a deep learning method under physical constraints as the physics-informed neural networks (PINNs), which makes the neural network an interpretable optimization problem. With the concept of automatic differentiation, spatial and temporal coordinates' partial derivatives can be conveniently calculated using the weights and bias of the network, and then are included in the loss function [17]. Additionally, this method has the advantages of being mesh free and being usable for solving PDEs regardless of the equations' structure or even their complexity [18]. Moreover, the training points can be randomly sampled from the domain of interest, and less training data samples are required, compared with data-driven methods, avoiding the curse of dimensionality. The trained neural network is an analytical approximation of the latent solution, and hence, no interpolation is required for the solution at new input points [1], and further computation of derivatives, such as gradients, is convenient and smooth. Raissi employed PINN to solve one-dimensional Burger partial differential equations and the inverse problem of 2D/3D PDEs. After that, research works based on the PINN mushroomed. Jagtap et al. [19,20] proposed a parallel calculation method, which decomposes the domain into a few subdomains, then uses separate networks to represent each subdomains' function. Song et al. [21] leveraged PINN to solve the frequency-domain wave equation and introduced an adaptive sinusoidal activation function to improve the training performance. Lu et al. [22] developed a Python toolkit based on Tensorflow, which is named DeepXDE and integrates the application of the PINN. Fang et al. [23] studied the three-dimensional time-independent surface problem. Hu et al. [24] researched the two-dimensional waveguide problem governed by the Helmholtz equation and estimated the unknown wave number with the help of PINN. Alkhalifah et al. [25] utilized PINN to predict the scattered wavefield function. Bin Waheed et. al. [26] used PINN to solve the nonlinear Eikonal equation and demonstrated the transfer learning possibilities of PINN. Guo et al. [27] verified PINN's ability in solving the wave equation, the KdV–Burgers equation, and the KdV equation. In addition, PINN was also applied to the uncertainty quantification problem [28,29] and the atomic simulation of materials [30].

Since the introduction of PINN, only few have worked on the computation of multiphysics problems [31]. In this article, we propose to use the PINN to solve a Joule heating problem, which consists of a coupling problem of the electrokinetic field and steady-state thermal field. The study is carried out with the help of the deep learning framework, PyTorch. To compute the electric field or the current density at a point of the domain, the calculation of the gradient of the state variable, i.e., the electric scalar potential, is needed. The automatic differential function of PyTorch makes the computation convenient for this solution. Therefore, PINN has the capability of handling the electro-thermal coupling problem. To explore the feasibility and effectiveness of this method, we employ two neural networks to approximate those two fields, namely the steady electric field and steady-state thermal field, and then combine them with a sequential coupling.

The rest of the paper is organized as follows. Section 2 recalls the idea behind the PINN framework and presents the method we use for applying PINN on electro-thermal analysis. Section 3 utilizes a concrete case to solve the electro-thermal coupling problem. Then, we compare the accuracy and continuity of the gradient field between the first-order FEM and the PINN. Section 4 discusses the experimental results and provides some prospects for future work. Section 5 concludes this work.

## 2. Materials and Methods

In this section, we first introduce the principle of the deep neural network (DNN), followed by an overview of PINN.

### 2.1. DNN

DNN is a form of multi-layer perceptrons, which can be interpreted as universal function approximators [32]. The DNN consists of $L$ layers and an input vector called the input layer, wherein $L$ layers include $L-1$ hidden layers, and the $L^{th}$ layer is the output layer. We let $N^L : R^{C_i} \to R^{C_o}$ denote a deep neural network of $L$ layers, where $C_i$ and $C_o$ denote dimensions of the input vector and output vector, respectively, and $N_k$ ($1 \le k \le L$) denotes neurons in $k^{th}$ layer ($N_L = C_o$). Let $\mathbf{W}^k \in R^{N_k \times N_{k-1}}$ and $\mathbf{b}^k \in R^{N_k}$ denote the weight matrix and bias vector of the $k^{th}$ layer, respectively. The activation function in the $k^{th}$ layer is denoted by $\sigma^k(\cdot)$, regardless of its type. Commonly used activation functions are Sigmoid, ReLu, Tanh, etc. After the $k^{th}$ hidden layer receives an input vector $\mathbf{z}^{k-1}$ ($\mathbf{z}^0 = \mathbf{x}$), it multiplies the vector with the weight matrix $\mathbf{W}^k$, and then adds it to the $\mathbf{b}^k$. If $k$ is equal to $L$, the result is output directly; otherwise, an activation function is applied to the result and then passed to the next layer. In summary, the feed-forward process of DNN is given in Equation (1). The first hidden layer's input vector is $\mathbf{x}$ and output is $\mathbf{z}^1$; the rest of the hidden layers will have an input $\mathbf{z}^{k-1}$ ($2 \le k \le L-1$) and output $\mathbf{z}^k$. For the regression task, the last layer usually does not require an activation function, and the $L^{th}$ layer takes in $\mathbf{z}^{L-1}$, and outputs $\mathbf{y}$, as follows.

$$
\begin{aligned}
\mathbf{z}^1 &= \sigma^1(\mathbf{W}^1\mathbf{x} + \mathbf{b}^1), \\
\mathbf{z}^2 &= \sigma^2(\mathbf{W}^2\mathbf{z}^1 + \mathbf{b}^2), \\
&\ \ \vdots \\
\mathbf{z}^{L-1} &= \sigma^{L-1}(\mathbf{W}^{L-1}\mathbf{z}^{L-2} + \mathbf{b}^{L-1}), \\
\mathbf{y} &= \mathbf{W}^L\mathbf{z}^{L-1} + \mathbf{b}^L.
\end{aligned}
\tag{1}
$$

Figure 1 shows the basic structure of DNN. In DNN, the hyperparameters are a set of parameters used to control the learning process, e.g., the number of hidden layers and the number of neurons in each layer.

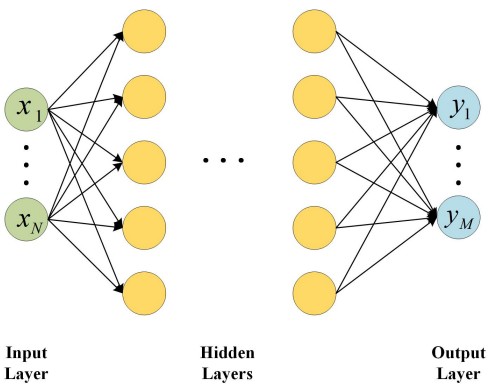

**Figure 1.** A basic schematic representation of DNN, which is composed of an input layer along with data $\mathbf{x} = [x_1, \ldots, x_N]^T \in R^N$, an output layer $\mathbf{y} = [y_1, \ldots, y_M]^T \in R^M$ and hidden layers.

### 2.2. PINN

Machine learning is able to act as a surrogate model, discovering the potential relations between the input and output, classifying various images, and predicting system responses under different conditions [33]. Most methods that are based on machine learning for solving PDEs are data driven and do not consider any physical constraints [27]. Hence, a great amount of data is often required to train the model. Preparing training data in advance is undoubtedly an arduous process. On the contrary, PINN introduces physical constraints and reformulates solving equations as an optimization problem, which reduces the amount of required data, alleviates overfitting issues, and therefore improves the robustness of the trained model [34].

The following describes the formulation of PINN for solving general PDEs. Considering the following PDE with a boundary condition

$$Lu(x) = f(x), x \in \Omega, \tag{2}$$

$$u(x_0)|_\Gamma = B(x_0), \Gamma \subset \partial\Omega. \tag{3}$$

where $u(x)$ denotes the solution to this PDE, $L(\cdot)$ is a differential operator, $f(x)$ is a forcing function, and $\Omega$ represents the domain of interest, $\partial\Omega$ is a symbol of all boundaries. $\Gamma$ denotes specific boundary where the boundary condition is imposed.

As mentioned previously, we can approximate the solution of PDEs by DNNs. The essence of PINN is to use the PDE as a loss function for the optimization. All NN training requires an optimization process. Figure 2 graphically introduces the structure of PINN. The input data $x$ is a set of randomly sampled points from the domain and boundaries. After obtaining the loss function value, we will judge both the loss value and the iteration value to determine whether to enter the next step. $\epsilon$ denotes the cut-off threshold of loss value, and *it* refers to the maximal iteration number. Once the loss value is less than the threshold $\epsilon$, or the iteration is greater than the maximal number *it*, then the training will end. With the help of PyTorch's automatic differentiation module, we can obtain the output's partial derivatives of any order. Consequently, it simplifies converting the network output into a governing equation format. We let $\Theta = \{\mathbf{W}^1, \mathbf{W}^2, \ldots, \mathbf{b}^1, \mathbf{b}^2, \ldots\} \in V$ be a set of trainable parameters, i.e., all the layers' weights $\mathbf{W}^k$ and biases $\mathbf{b}^k$ ($1 \leq k \leq L$), where $V$ denotes the parameter space. We also let $u(x; \Theta)$ denote the output function of PINN. The loss function $J(\Theta)$ can be defined as [16].

$$J(\Theta) = MSE_F + MSE_B, \tag{4}$$

$$MSE_F = \frac{1}{N_f} \sum_{i=1}^{N_f} |Lu(x_f^i; \Theta)^2, \tag{5}$$

$$MSE_B = \frac{1}{N_B} \sum_{i=1}^{N_B} |u^i - u(x_B^i; \Theta)|^2. \tag{6}$$

Equation (5) denotes the governing equation constraint, and Equation (6) is the boundary condition constraints, where $N_f$ and $N_B$ are the number of domain samples and boundary samples, respectively. $x_f^i$ denotes the sampling points in the domain, $u^i$ denotes the target outputs, and $x_B^i$ denotes the sampling points on the boundaries. Unless the value of $J(\Theta)$ converges to zero, we end up with an approximation of the PDE's solution. By adjusting the training parameters $\Theta$ to minimize the loss function, we seek to find the optimal parameters $\Theta^*$ that satisfy the following condition.

$$\Theta^* = \underset{\Theta \in V}{\arg\min} \, J(\Theta). \tag{7}$$

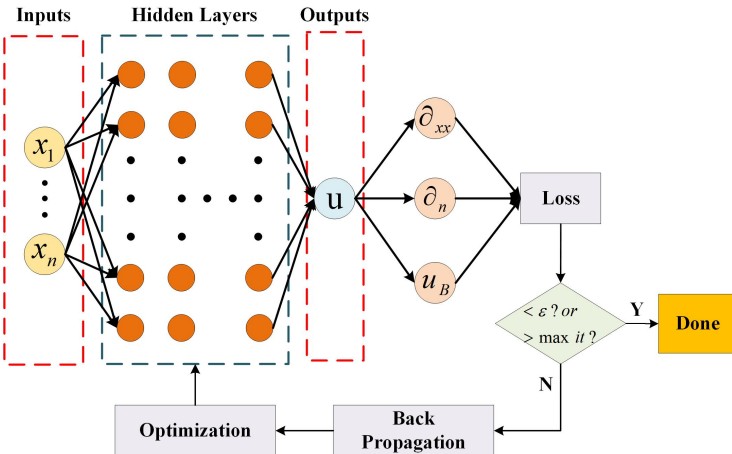

**Figure 2.** A schematic plot of PINN.

The optimization of the loss function depends on the backpropagation and optimizer. Common optimizers include SGD, Adam, and L-BFGS. Furthermore, to satisfy the boundary conditions, there currently are two methods, namely the "soft" and the "hard" boundary conditions. The "soft" boundary condition method is more commonly used since it embeds the boundary conditions into the loss function directly without recourse to preprocessing; "hard" constructs a particular solution function which is usually used to automatically satisfy the Dirichlet boundary conditions, thereby making the optimization more efficient [13]. In addition, it reduces the number of training points required because we only need to sample points from the inside of the domain and from the boundaries where the "soft" condition is applied, thus reducing the training cost [1,35]. The corresponding ansatz for the solution with a "hard" boundary condition is

$$\hat{u}(x; \mathbf{\Theta}) = G(x) + D(x)u(x; \mathbf{\Theta}). \tag{8}$$

where $G(x)$ denotes a smooth extension function, which satisfies the Dirichlet boundary condition, $D(x)$ is a smooth distance function that gives the distance from $x \in \Omega$ to $\Gamma$, and $\hat{u}(x; \mathbf{\Theta})$ represents the new output function designed to automatically fulfill the Dirichlet constraints based on the raw output $u(x; \mathbf{\Theta})$. Since the Dirichlet boundary can be automatically satisfied, it is no longer considered in the loss function. Nevertheless, it is still necessary to train for the Neumann boundary and Robin boundary terms by adding the boundary constraints to the loss function.

### 2.3. PINNs for the Electro-Thermal Coupling Problem

In this section, we introduce the principle of solving a two-dimensional electro-thermal coupling problem with PINNs. The electrokinetic field problem seeks to find the electric field or current density distribution in a conducting region. The problem can be described by the electric scalar potential in the form of the Laplace equation. Considering $u(x, y)$ as the space distribution of the electric potential, $\sigma$ denotes the electric conductivity, $\partial u / \partial \mathbf{n}$ denotes the normal derivative of $u$ along the outward direction of the Neumann boundary $\Gamma_N$, $B(x, y)$ is the given value of $u$ on the Dirichlet boundary $\Gamma_D$, and $u(x, y)$ satisfies the following partial differential equation and boundary conditions:

$$\nabla \cdot \sigma \nabla u(x, y) = 0, \tag{9}$$

$$\frac{\partial u(x_N, y_N)}{\partial \mathbf{n}}\Big|_{\Gamma_N} = 0, \tag{10}$$

$$u(x_D, y_D)|_{\Gamma_D} = B(x, y). \tag{11}$$

The electric current generates Joule heat, which causes the temperature change of the conductor. The calculation formula for Joule heat $Q$ at each point is

$$Q = \mathbf{J} \cdot \mathbf{E}. \tag{12}$$

where $\mathbf{J}$ denotes the electric current density and $\mathbf{E}$ the electric field, which is expressed as $\mathbf{E} = -\nabla u$. The constitutive equation is given by

$$\mathbf{J} = \sigma \mathbf{E}. \tag{13}$$

The thermal field induced by the Joule heat is in the form of steady-state heat conduction. Considering $T$ as the spatial temperature distribution, $k$ as the thermal conductivity, and $Q$ as representing the Joule heat source, which can be computed by Equations (12) and (13), the governing equation and boundary condition is given by

$$-\nabla \cdot k\nabla T(x, y) = Q, \tag{14}$$

$$T(x_D, y_D)|_{\Gamma_D} = T_0. \tag{15}$$

where $T_0$ is set to 273$K$.

To handle the electro-thermal coupling problem, we employ two PINN networks to approximate solutions separately and train the networks in a sequential manner. The flow chart is shown in Figure 3, both of the two neural networks' inputs are the same sampling points, and the outputs are the electric scalar potential and the temperature, denoted by $u$ and $T$, respectively. The corresponding loss functions of those two problems are listed below. Since we employ the "hard" boundary condition method here, the loss function for the Dirichlet boundary is eliminated. Equations (16) and (17) denote the PDE constraint and Neumann boundary constraint of the steady electric field problem, respectively. $\hat{u}(x_f^i, y_f^i; \Theta)$ denotes the output of PINN corresponding to the domain sample points, and $\hat{u}(x_N^i, y_N^i; \Theta)$ represents the output of the Neumann boundary sample points.

$$MSE_F = \frac{1}{N_f} \sum_{i=1}^{N_f} |\frac{\partial}{\partial x}(\sigma \frac{\partial \hat{u}(x_f^i, y_f^i; \Theta)}{\partial x}) + \frac{\partial}{\partial y}(\sigma \frac{\partial \hat{u}(x_f^i, y_f^i; \Theta)}{\partial y})|^2, \tag{16}$$

$$MSE_B = \frac{1}{N_B} \sum_{i=1}^{N_B} |\frac{\partial \hat{u}(x_N^i, y_N^i; \Theta)}{\partial \mathbf{n}}|^2. \tag{17}$$

Equation (18) is the PDE constraint of the thermal field problem. $\hat{T}(x_f^i, y_f^i; \Theta)$ is the output of PINN corresponding to the temperature domain sample points. No $MSE_B$ term is needed since the thermal field problem only has Dirichlet boundary conditions.

$$MSE_F = \frac{1}{N_f} \sum_{i=1}^{N_f} |(\frac{\partial}{\partial x}(k \frac{\partial \hat{T}(x_f^i, y_f^i; \Theta)}{\partial x}) + \frac{\partial}{\partial y}(k \frac{\partial \hat{T}(x_f^i, y_f^i; \Theta)}{\partial y}) + Q|^2. \tag{18}$$

Firstly, we train the neural network of the steady electric field problem, where the loss function is embedded with the governing equation and boundary conditions of Equations (9)–(11). When the loss function drops below a designated cut-off threshold or when the iteration reaches a specified number, the PINN training ends. Subsequently, we calculate the gradient of $u$ to obtain the distribution of the electric field and the current density and then compute the Joule heat source based on the former results. The heat source is passed to the next PINN network to approximate the thermal field Equations (14) and (15). It can be noticed that in both PINNs, the "hard" condition is applied for the Dirichlet boundary conditions. After the same flow of work, we finally obtain both the electric field and thermal field's distributions.

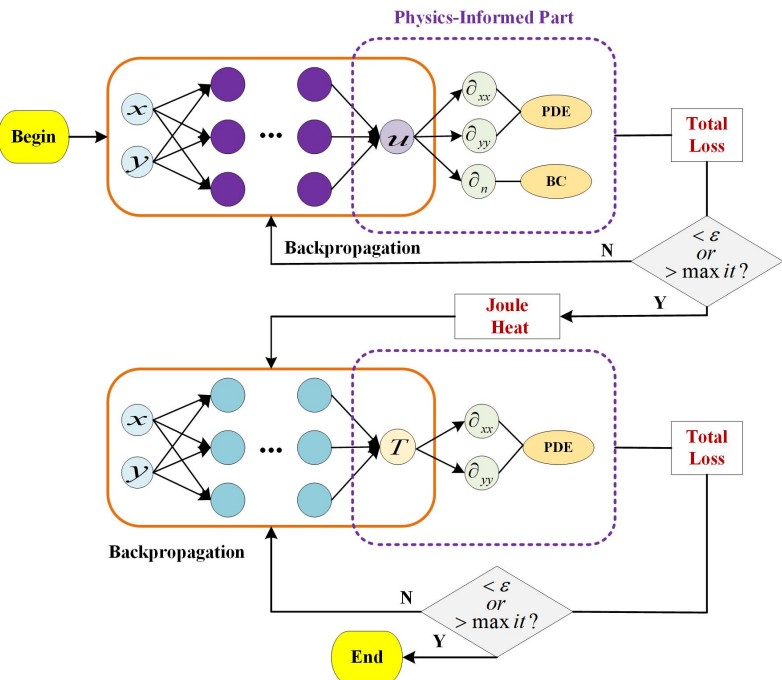

**Figure 3.** The flow diagram of electro-thermal coupling computation based on PINN.

## 3. Experiments and Results

### 3.1. Electro-Thermal Coupling Problem

In this section, the two deep neural networks whose loss functions are defined in Equation (4) are employed to study the electro-thermal coupling issue. The governing equations and boundary conditions of the electrokinetic problem are defined in Equations (9)–(11), and counterparts of the thermal problem are defined in Equations (14) and (15). The deep learning framework we choose is PyTorch. The configuration of the computer is Intel Core i7-9750H for CPU, Nvidia GTX1660Ti 6G for GPU, with 16 GB RAM. We use PyTorch's automatic differentiation [36] to establish the governing equations and the Neumann boundary constraints.

A Rectangle Electro-Thermal Coupling Problem

We choose a model of a square geometry $[-0.5, 0.5] \times [-0.5, 0.5]$ m². As a proof of concept, here, we take the electric conductivity and thermal conductivity as isotropic scalars, and we set $\sigma = 1$ S/m and $k = 1$ W/(m $\cdot$ K). The sample results are shown in Figure 4. There are 100 sample points in the domain for both the electric and thermal field problems. Each field has different boundary conditions; for the steady electric field, the upside border of the model is applied to a voltage of 1 V, while the downside border is grounded and the Neumann boundary condition is applied to the lateral lines. Since we use the "hard" boundary method to process the Dirichlet boundary condition, the upper and lower boundaries do not need to be sampled for the electric field problem. As a result, only 40 points on the two lateral boundaries are sampled for the electric field problem. While for the thermal field, all the boundaries are Dirichlet boundaries, their boundary conditions are also automatically satisfied by the "hard" boundary method, and hence, there is no need to sample on these boundaries. The points in the inner region are sampled by the Latin hypercube sampling method [37], and the boundary data are sampled randomly in a uniform distribution.

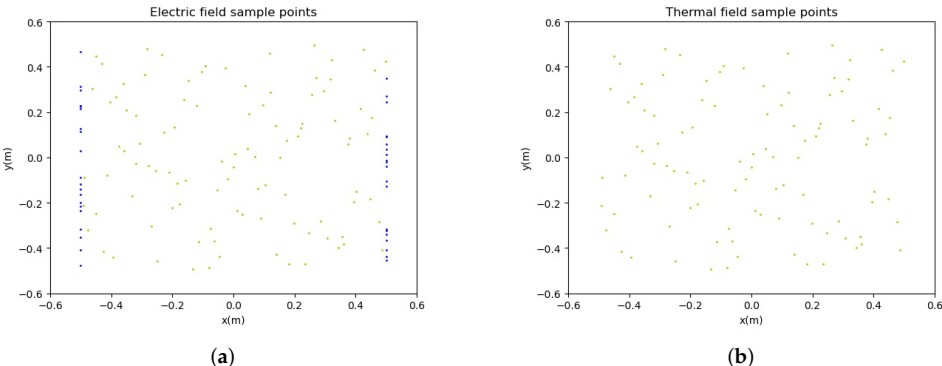

**Figure 4.** Sample points: (**a**) for the electric field problem; (**b**) for the thermal field problem.

For our study, the electrokinetic field's governing equation is the Laplace equation. Here, we construct a "hard" boundary model adapted to the geometry and boundary conditions. The smooth extension function $G(x,y)$ and the smooth distance function $D(x,y)$ defined in Equation (8) are specifically defined as follows, respectively.

$$G(x,y) = y + 0.5, \tag{19}$$

$$D(x,y) = (0.5^2 - y^2). \tag{20}$$

After the "hard" boundary condition is applied, since we treat the electric conductivity as isotropic scalar, the final PDE expressions of the electric field problem based on the raw NN output $u(x,y;\Theta)$ are given by

$$-\sigma(u\nabla^2 D + 2\nabla D\nabla u + D\nabla^2 u + \nabla^2 G) = 0. \tag{21}$$

Since the Dirichlet boundary is compulsorily satisfied, the loss term of the steady electric field network model only includes governing equations and constraints of the Neumann boundaries.

When the training of the electric field is over, we can obtain a satisfactory electric field solution; then, the Joule heat is calculated according to the obtained result. Next, the heat source data are transferred to the thermal field to continue the calculation. For the heat conduction equation, we also construct "hard" boundary conditions. Since we set the temperature on the boundaries to 273 $K$, we have $G(x,y) = 273$. The distance function is given by

$$D(x,y) = (x^2 - 0.5^2)(y^2 - 0.5^2). \tag{22}$$

Similarly, the final PDE expression of the thermal field problem based on the raw NN output $T(x,y;\Theta)$ is given by

$$-k(T\nabla^2 D + 2\nabla D\nabla T + D\nabla^2 T + \nabla^2 G) = Q. \tag{23}$$

Two separate PINNs are employed to approximate the steady electric field and the thermal field, respectively. After parameter tuning of the neural network, we can obtain a well-trained model. For both neural networks, we choose 6 hidden layers with 20 neurons in each layer. The activation function we select is Tanh, the optimization algorithm is Adam [38], and the initial learning rate of both networks is $lr = 1 \times 10^{-2}$. For training data, we utilize full batch size. Moreover, we set the cut-off threshold of the loss function as $1 \times 10^{-5}$ and the maximum number of epochs at 20,000, that is, when the neural network training meets any of these two conditions, the training ends. Weights and biases of the neural network usually need to be initialized by some initialization methods. Here, we choose a widely used method: Xavier initialization [39]. Based on the experiment results, the number of epochs need for the two field problems are 1108 for the steady electric field,

and 1182 for the heat conduction field. The lowest loss function values and the spent time of each neural network training are displayed in Table 1. We can notice that because of the non-homogeneous distribution of the thermal field, the learning of the temperature distribution is more difficult than the electric field.

**Table 1.** The lowest loss values and the time consumed in each training.

| Property | Steady Electric Field | Thermal Field |
| --- | --- | --- |
| Lowest Loss | $9.97 \times 10^{-6}$ | $9.84 \times 10^{-6}$ |
| Time(s) | 101 | 109 |

Our training results are shown in Figure 5. We use the FEM results as the reference solutions to evaluate the training accuracy. The FEM results, shown in Figure 6, are obtained with a mesh of 6592 points and the CPU time of 0.87 s.

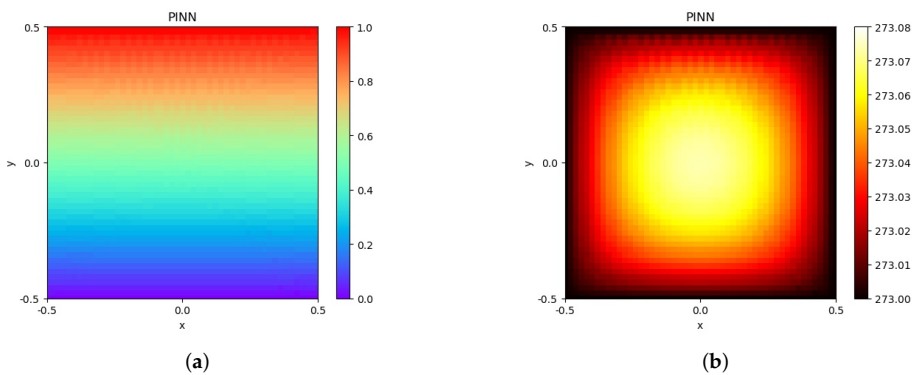

(a)                                      (b)

**Figure 5.** Results of our PINN networks: (**a**) the electric potential (V); (**b**) the temperature (K).

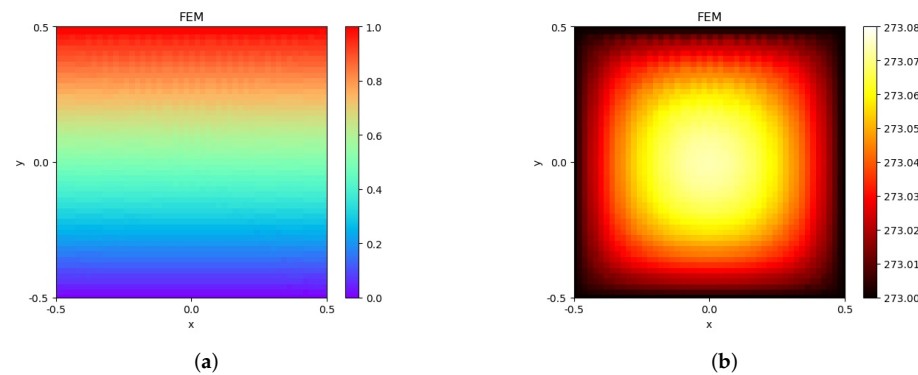

(a)                                      (b)

**Figure 6.** Result of the FEM: (**a**) the electric potential (V); (**b**) the temperature (K).

In order to further compare the difference between our training results and the reference solutions, Figure 7 shows the absolute error distributions between the PINNs solutions and the reference solutions. The relative errors are below $4 \times 10^{-4}$ for both problems. We can conclude that DNN's training results are in good agreement with those calculated by FEM; however, the large computation time of DNN as compared to that of FEM is an apparent shortcoming that needs to be improved.

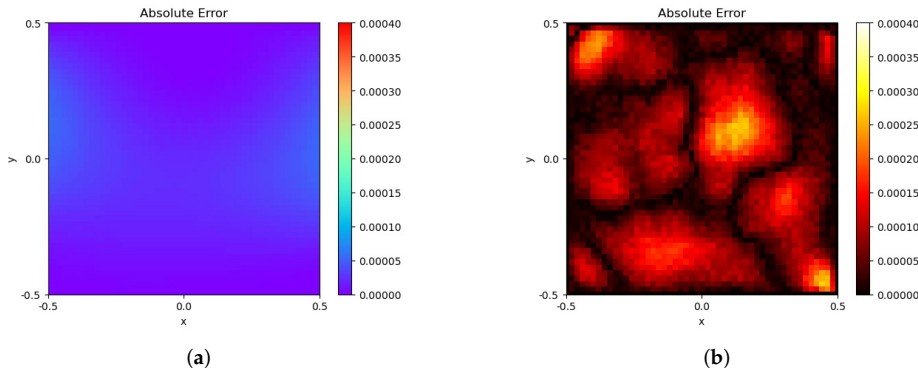

**Figure 7.** The absolute error between the PINN result and the FEM results: (**a**) the electric potential (V); (**b**) the temperature (K).

### 3.2. The Comparison between FEM and PINN

The trained PINN could be viewed as an analytical approximation of the latent solution; hence, for further exploration, such as the derivatives of state variables such as the gradient manipulation, the PINN can do this analytically and easily [1]. This advantage makes post-processing operations convenient. In addition, PINN does not need to be retrained to solve the solution and its gradient at the new sampling points. Hence, we can obtain a smooth derivative distribution. However, for FEM, the differential equations can only be solved at discrete points; when utilizing the first-order FEM to compute the state variables' gradients, such as the field strength, the value inside the element is constant and discontinuous at the elements' interfaces. Therefore, for FEM, the solution state variables' gradients are less accurate.

To compare the accuracy and the continuity of the gradient field calculated by FEM and by PINN, respectively, we choose an equation, which has an analytical solution. The research domain is a square of $[0, 1] \times [0, 1]\ m^2$, and the PDE is given by

$$-\nabla^2 u = 2\pi^2 \sin \pi x \sin \pi y, \tag{24}$$

$$u|_{\partial\Omega} = 0. \tag{25}$$

the analytical solution is

$$u(x, y) = \sin \pi x \sin \pi y. \tag{26}$$

We first discretize the model, acquire the solutions of the scalar potential on the nodes by first-order FEM, and then calculate the gradient inside every element. The mesh diagram is shown in Figure 8a, where the number of grid nodes is 983. In order to facilitate the comparison and the illustration, we compute the gradient module value on the line $y = 0.2$ as illustrated by the horizontal solid line in the picture. Figure 8b shows the gradient modulus comparison between the first-order FEM and the analytical solution. It can be seen that for the FEM, the gradient modulus distribution of the scalar potential is discontinuous along the line; this is because the gradient modulus obtained by the first-order FEM is constant per element.

Secondly, we utilize PINN to solve the equation. After tuning the deep learning parameters, we choose the number of sample points to be 1000, the number of layers to be 2, and for there to be 60 neurons for each hidden layer. The activation function and optimization function we select are the same as in the previous section, while the learning rate is $lr = 5 \times 10^{-3}$. We set the maximum number of epochs as $20,000$ and the cut-off threshold for the loss function as $1 \times 10^{-3}$. Finally, the training time is 78 s, the training takes 1166 epochs, and the lowest loss value is $9.98 \times 10^{-4}$. After training the model, we resample points on the $y = 0.2$ line and calculate their solution, gradients, and the corresponding gradient modulus. Figure 9 illustrates the gradient modulus comparison

between the PINN and the analytical solution. It can be seen that the result of PINN is in good agreement with the analytical solution, given by a smooth distribution curve.

However, FEM solves this problem with a mesh of 983 points with a computational time of merely 0.12 s. Compared with FEM, the large computational time is still a limitation of PINN for now.

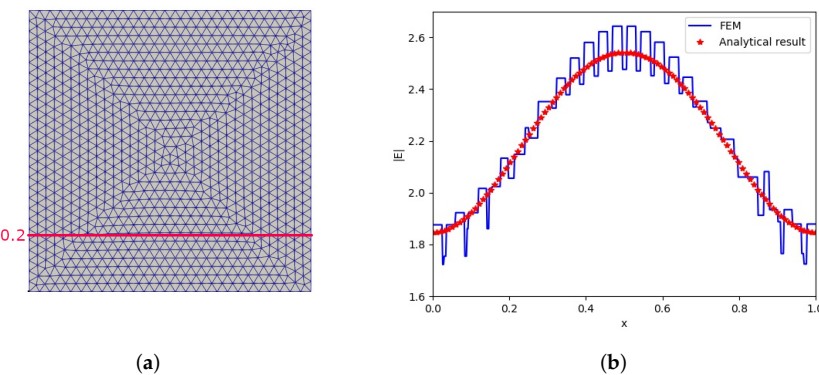

(**a**)                                                                                          (**b**)

**Figure 8.** (**a**) The mesh diagram of the model; (**b**) the gradient modulus comparison between the first-order FEM and the analytical result.

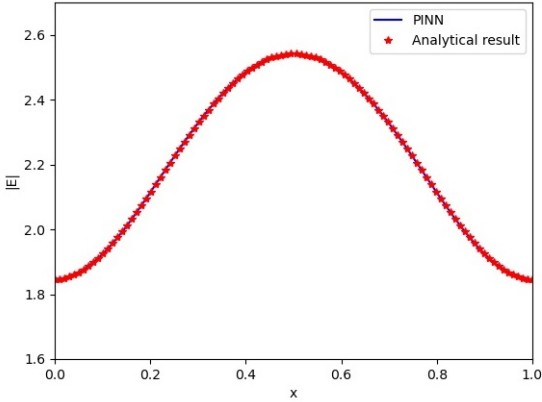

**Figure 9.** The gradient modulus comparison between the PINN result and the analytical result.

### 3.3. Empirical Properties of the PINN

For the problem defined by Equations (24) and (25), which has an analytical solution, we performed a study to quantify the training time and the predictive error for two different variables: one is the network's size and the other is the number of samples. Some empirical conclusions will be drawn. Specifically, the neural network's size includes the network's width and depth; the width refers to the number of each layer's neurons; and the depth refers to total number of hidden layers. In the following, we will study each of these two variables while the other variable value is set at a reasonable value. Hyperparameters that are not studied in this work are set to be the same as in the previous section. For each combination, we set the maximum epoch to $20,000$ and the cut-off threshold for the loss function to $1 \times 10^{-3}$. The root mean squared error (RMSE) is used to evaluate the prediction error. Considering $u_i$ as the analytical solution, $\hat{u}_i$ as the neural network's predictive value, and $n$ as the number of test points, the RMSE is given by

$$RMSE(u, \hat{u}) = \sqrt{\frac{1}{n} \sum_{i=1}^{n} (u_i - \hat{u}_i)^2}. \tag{27}$$

### 3.3.1. The Effect of Depth and Width of the Network on the Training Time

The selection of hyperparameters plays a vital role in the convergence time of PINN. In this section, we explore the influence of the network's size, i.e., the depth and the width of the network, on the training time. We fix the the number of samples in the domain to 1000, and choose different sizes of the neural network. The practical epoch and the training time of each network are shown in Tables 2 and 3, respectively. According to the results in Table 2, we observe that except for the first set, i.e., one hidden layer with 5 neurons reaches the maximum epoch before reaching the cut-off threshold, other combinations' loss functions converge below the cut-off threshold within the maximum epoch. Furthermore, it is worth mentioning that the prediction errors, i.e., RMSEs, of all selections of the size are below $5 \times 10^{-4}$, which satisfies our requirement. Moreover, based on the Table 3, we can conclude that as the size of the neural network increases, the convergence speed gradually increases at the beginning; however, when it reaches a certain degree, then by increasing the size, the improvement in computation time stagnates or even becomes worse. For the considered example, the best result is obtained for 2 layers and 60 neurons.

**Table 2.** The practical epoch for different neural network sizes.

| Depth<br>Width | 1 | 2 | 4 | 6 |
|---|---|---|---|---|
| 5 | 20,000 | 8539 | 5126 | 5193 |
| 10 | 13,120 | 4832 | 2634 | 3046 |
| 20 | 8791 | 2239 | 1430 | 1591 |
| 30 | 7493 | 1993 | 1146 | 1584 |
| 60 | 5958 | 1161 | 1343 | 2936 |

**Table 3.** The training time (s) for different neural network sizes.

| Depth<br>Width | 1 | 2 | 4 | 6 |
|---|---|---|---|---|
| 5 | 1129 | 547 | 410 | 497 |
| 10 | 739 | 311 | 211 | 293 |
| 20 | 495 | 144 | 115 | 153 |
| 30 | 422 | 128 | 92 | 152 |
| 60 | 335 | 75 | 108 | 286 |

### 3.3.2. Effect of Number of Samples on the Prediction Error and Training Time

In this section, we investigate the influence of the number of samples on the training time and prediction error of the neural network. We test various number of samples. With respect to the size of the neural network, we use 2 hidden layers with 60 neurons in each layer.

Table 4 displays the training time, RMSEs of predictive results as compared to the analytical solution, and the practical epochs for this experiment. We can see that all combinations' loss functions converge below the cut-off threshold within the maximum epoch. The Xavier initialization function sets different initial values for the weight matrix of the model, which makes the training results slightly different each time. Consequently, we can see that the training time corresponding to 1000 points in Table 4 is slightly different from the training results of the neural network with 2 hidden layers and 60 neurons in each layer in Tables 2 and 3.

Finally, we observe that too few sample points, such as 10, converge the fastest while resulting in a high prediction error. Other than this option, the convergence speed increases as the sampling number goes up at the beginning; however, when it reaches a certain level, the improvement of the convergence speed stagnates or even becomes worse.

**Table 4.** The training time (s) and prediction error RMSEs, and the number of epochs for various amount of samples used in the training.

| Number<br>Property | 10 | 100 | 500 | 1000 | 5000 | 10,000 |
|---|---|---|---|---|---|---|
| Time (s) | 35 | 97 | 91 | 74 | 97 | 101 |
| RMSE | $1.59 \times 10^{-1}$ | $5.60 \times 10^{-4}$ | $1.09 \times 10^{-4}$ | $5.14 \times 10^{-5}$ | $7.07 \times 10^{-5}$ | $5.48 \times 10^{-5}$ |
| Epoch | 551 | 1508 | 1123 | 1066 | 1480 | 1539 |

## 4. Discussions

From these studies, we can summarize the following observations.

- PINN embeds physical constraints into the loss function of the neural network by using automatic differentiation. The imposition of the "hard" boundary makes the approximate solution automatically meet the Dirichlet boundary, which accelerates the convergence speed and improves the prediction accuracy [1].
- In addition to the advantage of being mesh-free, PINN can also generalize the construction process of various PDEs. On top of that, classical methods, such as FEM, can only obtain the solution on discrete points, while further interpolation is required for other points. This property makes the solution of the state variables at the interpolated points less accurate. For PINN, when considering the points that do not appear in the training set, there is no need to conduct an interpolation scheme to obtain the solutions.
- With the help of automatic differentiation, the derivative of each state variable can be easily calculated. Hence, the derivative distribution is smooth. However, for the first-order FEM, the first derivative of the state variable inside of an element is constant, which makes the derivative distribution discontinuous.
- Based on the experimental results, the convergence speed of PINN gradually increases as the size of the neural network goes up at the beginning. However, when it reaches a certain point, the improvement in computation time stagnates or even becomes worse. In addition, when studying the number of training samples influence on the convergence speed, except the case where the sampling points are too few, the convergence speed increases as the number of training samples increases; however, when it reaches a certain number, the improvement in convergence speed stagnates or even becomes worse.
- According to our experiments, although the PINN offers unique advantages for solving PDEs, its computational efficiency is an obvious disadvantage compared to FEM. Therefore, figuring out how to accelerate the training of PINN is an important research topic. Ref. [40] introduced an efficient approach based on adaptive sampling strategy, which speeds up the computation of the PINN. In [19], parallel calculation of the PINN was successfully implemented, which can easily handle any complex regional problems, but the improvement in computing time is still on the way. Huang et al. [41] realized speeding up convergence for high-frequency wavefields solutions by using the information from a pre-trained model instead of initializing the PINN randomly.
- PINN can be conveniently utilized to generate a surrogate model in the parametric analysis. In [42], the authors conducted a sensitivity analysis experiment. They first trained the PINN with merely a few values of a specific parameter and then utilized the trained neural network to predict the solution to this parameter within a range. The result is less accurate but still useful for the specific condition, which shows the possibility of PINN in tackling this kind of issue; we will work on this subject in our future study.

## 5. Conclusions

In this article, PINN is applied to solve an electro-thermal coupling problem. Two networks are trained to approximate each field. The coupling process is sequential, with the

electric field being evaluated first, followed by the thermal field after the heat sources from the electric field are given. In order to speed up the training and improve the training accuracy, we employ the "hard" boundary method. The numerical results show the feasibility of PINN in tackling this kind of problem. A well-trained PINN can provide a global and smooth approximation of the state variable, which is convenient for the evaluation of derivatives. However, compared with FEM, the long computation time is still an obvious disadvantage of PINN and needs to be further addressed. Since the PINN was successfully used to generate surrogate models, we will work on the parametric analysis based on it in the future.

**Author Contributions:** Conceptualization, Z.R. and Y.M.; methodology, Y.M., Z.R. and S.Y.; software, Y.M.; validation, Z.R., X.X., and S.Y.; formal analysis, Y.M., Z.R., S.Y.; investigation, Y.M. and S.Y.; resources, Z.R., X.X., S.Y.; data curation, Z.R.; writing—original draft preparation, Y.M.; writing—review and editing, Z.R., X.X., and S.Y.; visualization, Y.M. and S.Y.; supervision, Z.R., X.X., and S.Y.; project administration, Z.R., X.X., and S.Y.; funding acquisition, Z.R., X.X. and S.Y.. All authors have read and agreed to the published version of the manuscript.

**Funding:** This research was funded by The Institute of Electrical Engineering, CAS (Y960211CS3, E155620101, E139620101).

**Institutional Review Board Statement:** No applicable.

**Informed Consent Statement:** No applicable.

**Data Availability Statement:** The study did not report any data.

**Conflicts of Interest:** The authors declare no conflict of interest.

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
