# Peer review of "A Preliminary Study on the Resolution of Electro-Thermal Multi-Physics Coupling Problem Using Physics-Informed Neural Network (PINN)"

_algorithms, doi:10.3390/a15020053_

Round 1
Reviewer 1 Report
In this paper, the authors present a method for solving electro-termal coupling problems using Physics-Informed Neural Network. The proposed method considered solving the time-dependent and static electro-termal coupling problems. The electro-termal coupling problems can be written as an coupling systems with one electrical equation and a thermal equation where the equations can be solved sequentially. In the proposed network the author consider a network is a composition of two sub-networks and each sub-networks represent a different equations. The sub-network is then trained sequentially to approximate the electric field and the thermal field, respectively. Numerical result are provided to demonstrate the performance of the proposed method.
This application of PINN is interesting and the paper is well structured. I recommend the paper for publication in Algorithms with minor revision.
The following items are some comments on this paper.
1. In equations (2) - (3), the author defined a partial differential equation and in equations (5)-(6), the authors introduced the loss function of the PINN. However, the authors have not stated the relation between the loss function and the PDE clearly. I suggest the authors can give a example to demonstrate How the loss function related to the PDE.
2. A followup question, in equation (4)-(6), since the loss function J is depending on $\Theta$. I suggest the author also indicate that the network $u$ is depend on $x$ and $Theta$ instead of just depend on $x$.
3. Although it may not be difficult to understand, I think it will be clearer to write down the loss functions for the sequential sub-networks.
4. In comparison between FEM and PINN, I think it is a little bit unfair. The FEM does not work well. I am not sure if it is because of the first degree finite element approximation. The approximation may be smoother if we use second degree finite element approximate or mixed finite element.

Reviewer 2 Report
The manuscript offers an approach to utilize physics-informed neural networks (PINNs) to solve electro-thermal coupling problems, and they utilize the hard constraint to avoid including the Dirichlet boundary condition as a loss term. Though the problem they solve is simple, the paper provides interesting insights, and the analysis the authors perform is helpful. However, the paper, though structured well, includes clarity gaps, mistakes, and inaccurate statements. I point them out in the uploaded annotated version of the paper. For example, in Figure 3, in the second network, the authors use u_B as boundary instead of T_B and include the neumann boundary derivative where it is not computed. Also, why not draw the Figure with the hard constraint included? This Figure as it stands does not represent their implementation. With respect to the training, it is not clear what the batch size is. Do the authors use a full batch. Also, in equations 17 and 18, why is D set to be negative once, and positive the other, respectively. Also, how is G determined in equation 16, as the boundary condition is not stated for the problem. Also, the authors should share the PDE’s makeup after utilizing the hard constraint.
With respect to the writing style, I made considerable corrections that I hope the authors incorporate. Including that all equations are considered sentences ending in a full stop or comma. I also recommend that the authors add a conclusion section to summarize key results.
Finally, I include additional suggestions, corrections, and comments in the uploaded annotated version of the paper. I encourage the authors to revise the paper accordingly. I wish them the best of luck in doing so.
Tariq Alkhalifah

Reviewer 3 Report
After reviewing your article, the overall performance is impressive. Some redundant words or sentences must be revised. As you have used the abbreviation at the first time, you don’t need to repeat it again. For instance, in Lines 2 and 3 are similar to Lines 15 and 83. In Lines 54 to 60, it’s a long sentence, which the readers can’t understand well. In Line 80, “give” should be “gives”. The 1e-3 should be 1x10-3 in Line 229. Please use the formal expression. The grammar and typo in content must be checked again.
Finally, we didn’t see the conclusion part. You should add it.
In technology concerns, I have some concerns.
- In section 2.1, how do you define bk or give this b in the kth layer? It’s not clear in the statement.
- In Eq.(13), the conductivity s maybe can be treated as a tensor, not a scalar. If yes, the matrix and calculation must be revised.
- In Line 146, the area unit is m2. If yes, this topic talks about the electro-thermal coupling issue should be dug into more. In the IC, the front-end device and the back-end line conduction are different. The device size is gradually narrowed down, especially entering the nano-node scale. Thus, the electro-thermal coupling issues with PINN are not easy to cover all of aspects. Please give us some solid comments.
- In Figs. 8 and 9, are the analytical results the true experimental consequences? If yes, it’s great. If not, the comparison between the analytical results and PINN is not richly meaningful. You should use a physical data or some published data to support your PINN model. In abstract, you mentioned the experimental results in Line 9. If yes, please also describe the experimental sensing a little.
Round 2
Reviewer 2 Report
The manuscript has improved and the authors have addressed my comments and suggestions in the revised version of the paper. I have minor language edits in the annotated uploaded version of the paper that the author can correct as they submit the final version for publication as the associate editor deems acceptable.
